# Cost-effectiveness of one-year adjuvant trastuzumab therapy in treatment for early-stage breast cancer patients with HER2+ in Vietnam

**Anh Quynh Nguyen**[1], **Oanh Thi Mai Tran**[2], **Phuong Khanh Nguyen**[2], **Ha Thu Nguyen**[1]*

**1** Department of Health Policy and Economics, Hanoi University of Public Health, Hanoi, Vietnam, **2** Vietnam Strategy and Planning Institute, Hanoi, Vietnam

* nth11@huph.edu.vn, nth11huph@gmail.com

## Abstract

**Data Availability Statement:** All relevant data are within the paper and its Supporting Information files.

### Background

In Vietnam, trastuzumab is included in social health insurance's benefits package with a reimbursement rate of 60%, but policymakers have been concerned about its cost-effectiveness. The research aims to evaluate the cost-effectiveness of one-year adjuvant trastuzumab therapy for early-stage breast cancer patients with human epidermal growth receptor 2 (HER2+) from a societal perspective.

### Method

A Markov model was developed and validated to estimate the lifetime cost and effectiveness (using life year and quality-adjusted life year) of one-year adjuvant trastuzumab therapy compared to chemotherapy (using paclitaxel) alone. Treatment efficacy and transition probabilities were estimated based on published trials (i.e., N9831, NSABP B-31, HERA, and BCIRG 006). Local cost and utility data were employed to capture the Vietnam context. One-way sensitivity analysis, probabilistic sensitivity analysis, threshold, and scenario analysis were also performed.

### Results

One-year adjuvant trastuzumab therapy combined with chemotherapy compared to chemotherapy alone yielded an additional cost of 888,453,971VND (39,062 US$) with an additional 3.09 LYs and 1.61 QALYs, resulting in an ICER of 287,390,682 VND (12,635 US$) per LY gained, or 519,616,972 VND (22,845 US$) per QALY gained. The ICER exceeds the cost-effective threshold of 1- and 3-time GDP per capita by 6.3 and 2.1 times. The probabilistic sensitivity analysis shows similar results. According to one-way sensitivity analysis, ICERs were driven mainly by transition probabilities and trastuzumab price. One-year adjuvant trastuzumab therapy would be cost-effective at the 3-time GDP per capita threshold if the cost of Herceptin 150mg and 450mg vials were reduced by 56% and 54%, correspondingly.

**Funding:** The development of the Markov model employed in this study was initiated from the Vietnam National Health Technology Assessment project period 2014-2018. The project was coordinated by the Vietnam Health Strategy and Policy Institute (HSPI), a unit under the Vietnam Ministry of Health responsible for providing evidence and consulting for the health policy-making process in Vietnam. The topic was selected based on the topic selection process for the Vietnam HTA project conducted by the HSPI. The funders had no role in study design, data collection and analysis, decision to publish, or preparation of the manuscript.

**Competing interests:** The authors have declared that no competing interests exist.

## Conclusion

In Vietnam, one-year adjuvant trastuzumab therapy for early-stage breast cancer with HER2+ is not cost-effective. The research provided reliable and updated evidence to support policymakers in revising the health insurance benefit package. The policymakers should consider the options to reduce the cost of trastuzumab (e.g., regarding the use of trastuzumab biosimilars, price negotiation options, and options of optimizing the use of Herceptin vials among concurrent hospitalized breast cancer patients).

## 1. Introduction

Breast cancer is the most common cancer worldwide [1] and the most commonly diagnosed cancer among women in Vietnam in 2020 [2]. The rapid increase of breast cancer incidence is associated with changes in the prevalences of established risk factors (i.e., early age at menarche, later age at menopause, advanced age at first birth, fewer number of children, less breastfeeding, menopausal hormone replacement therapy, oral contraceptives, alcohol consumption, excess body weight, and physical inactivity) parallel with increased detection through organized and opportunistic breast cancer screening [3]. Vietnam has experienced a similar trend of some related risk factors (i.e., alcohol consumption, body mass index) [4] and probably the upward trend in the uptake rate of breast cancer screening [5]. In response to the growing global breast cancer burden and particularly the premature mortality in transitioning countries, the Global Breast Cancer Initiative (launched by the WHO and international partners in 2021) emphasizes the importance of increasing access to breast cancer early diagnosis and prompt, comprehensive cancer management [6]. Providing immediate and effective treatment for patients diagnosed with breast cancer is extremely important in the context of a rapid increase in incidence cases.

Trastuzumab is a monoclonal antibody that manages human epidermal growth factor receptor 2 (HER2+) breast cancer. The HER2 gene is positive in about 20% of primary invasive breast cancer [7]. It is well known that HER2-positive breast cancer is associated with poorer outcomes and higher mortality rates than other breast cancer subtypes. Trastuzumab, in combination with chemotherapy as adjuvant therapy in HER2+ early-stage breast cancer, is widely recommended due to its effectiveness in increasing overall survival and disease-free survival [8]. Recommendation on using one-year trastuzumab treatment as adjuvant therapy for HER2+ early-stage breast cancer is consensus in both international guidelines [9,10] and Vietnam national guideline [11].

In Vietnam, the primary sources of health financing are the state budget and the Social Health Insurance (SHI) fund. The state budget plays a pivotal role in funding the preventive sector and non-autonomous hospitals. Meanwhile, the SHI, which achieved a remarkable 91% coverage rate in 2021, plays a fundamental role in financing curative care. The SHI's benefits package is primarily focused on curative services, including select primary healthcare services. Fee-for-service serves as the dominant provider payment mechanism under the SHI scheme, but it often falls short of covering the entire direct healthcare cost. Regarding medications, the Ministry of Health (MOH) periodically updates the list of medicines eligible for SHI reimbursement every two to three years, along with the corresponding reimbursement rates. Currently, trastuzumab is part of the SHI benefits package with a reimbursement rate of 60% [12]. For other commonly prescribed medications and curative services, co-payment rates are set at either 0% (for specific groups, such as children under 6, individuals with disabilities,

impoverished households, and ethnic communities), 5% (for those receiving social allowances, near-poor households, etc.), and 20% (for all other beneficiaries). Out-of-pocket payments continue to constitute a significant portion of total healthcare expenditures, exceeding 40%.

According to the Vietnam national treatment guideline for the breast cancer [11], one-year adjuvant trastuzumab therapy is recommended for patients with HER2+ early breast cancer. As mentioned above, although trastuzumab is included in the SHI benefits package with a reimbursement rate of 60% [12], policymakers have been concerned about its cost-effectiveness due to its high cost. While trastuzumab is indicated to be cost-effective in many Asian countries such as China, Iran, Japan, Singapore, and Taiwan [13], it is not cost-effective in many other settings with higher income [14]. Thus, this study aims to evaluate the cost-effectiveness of adjuvant trastuzumab in combination with standard chemotherapy compared with standard chemotherapy alone in the Vietnam context from the societal perspective. The study will provide valuable evidence for the local authority in deciding whether to include trastuzumab in the social health insurance benefits packages.

## 2. Methods

We conducted a model-based economic evaluation in compliance with the Vietnam health technology assessment guideline [15]. The study protocol was developed and consulted with experts from the Vietnam National Strategy and Policy Institute (HSPI) and Thai experts from the Health Intervention and Technology Assessment Program (HiTAP). A Markov model over a lifetime horizon (with one year cycle) was constructed from societal perspectives to compare the cost and outcome of the two interventions. The societal perspective was chosen to fully cover the impact of the intervention on Vietnam society since the out-of-pocket payment among breast cancer patients remains very high [16]. We followed the Consolidated Health Economic Evaluation Reporting Standards 2022 (CHEERS 2022) for reporting the study [17]. See Electronic Supplement (S1 Table) for the completed CHEERS checklist.

### 2.1. Intervention and comparator

We compared the costs and effectiveness of using one-year adjuvant trastuzumab combined with chemotherapy versus chemotherapy alone for Vietnamese women with HER2-positive early-stage breast cancer. The topic was prioritized based on a topic selection process for health technology assessment (HTA) conducted by the HSPI with consultation from HiTAP [18]. Trastuzumab (8 mg/kg followed by 6 mg/kg) is administered once every three weeks. This three-weekly injection is the most common administration method for HER2+ breast cancer patients. The duration of trastuzumab treatment is one year since it is the standard treatment based on the national breast cancer treatment guideline approved by the MOH [11]. The chemotherapy agent chosen for analysis is paclitaxel. According to expert opinions, paclitaxel is the most frequently prescribed chemotherapy agent among recommended early-stage breast cancer chemotherapies. Among other chemotherapy agents for breast cancer, paclitaxel also had the highest reimbursed expenditure in 2016.

### 2.2. Study population and setting

The hypothesized population cohort consists of 5,052 Vietnamese women eligible for trastuzumab treatment, i.e., patients with HER2+ early-stage breast cancer who qualified for the baseline cardiac assessment before trastuzumab treatment. The total number of eligible entering the model was estimated as follows. Step 1: we multiplied the total number of breast cancer patients in Vietnam in 2020 (i.e., roughly 60,700 patients) by the rate of breast cancer patients diagnosed in the early stage (i.e., 66%) and the proportion of early-stage breast cancer patients

having the results of HER2+ (i.e., 20%) to estimate the total number of early-stage breast cancer with HER2+ [1]. Step 2: we applied the result from Step 1 with the treatment coverage (i.e., 70%). The treatment coverage was estimated based on expert opinion considering that trastuzumab is only reimbursed for insured patients at the central level of general hospitals and provincial oncology hospitals. Step 3: we adjusted the result from step 2 with the proportion of patients passing the baseline cardiac assessment before trastuzumab treatment (i.e., 90%) to estimate the size of the population entering the model.

All patients enter the model at 50 years old, similar to the mean age of patients in different randomized control trials [19,20] on the effectiveness of trastuzumab. According to experts' opinions and reports on the mean age of breast cancer patients in previous studies [16,21–23], the age of 50 also reflects the onset age at which the study population starts the trastuzumab treatment in Vietnam. The average weight of the patient was 52.9 kg based on an analysis of 376 breast cancer patients in a quality-of-life survey conducted in three oncology hospitals across Vietnam in 2015 [23] and 256 breast cancer patients in a cost-of-illness study undertaken in four oncology hospitals across Vietnam in 2014 [16].

## 2.3. Model development and validation

A Markov model was developed for HER+ early-stage breast cancer in Vietnam women. The model was initially constructed based on reviewing the current national treatment guideline [11], similar economic evaluations [24–26], and treatment outcomes reported in related RCTs [27–29]. Then, the model structure was validated by clinical experts to reflect current management approaches for HER2+ breast cancer patients in the Vietnam healthcare settings. We conducted two expert panels and six in-depth interviews to gather opinions from experts. The first expert panel (which included five health economists from HSPI and five health economists from HiTAP) focused on face validation in terms of problem formulation (i.e., to determine whether the setting, population, interventions, outcomes, assumptions, and time horizons correspond to research topic) and model structure (to determine whether the model includes all possible health states). The second expert panel (which included three oncologists, two experts from VSS, three public health specialists, five health economists, two pharmacists, and one patient representing the breast cancer patient group) focused on face validation in terms of model structure and data sources (i.e., to identify the best available data sources for use). In-depth interviews with six key informants (including three oncologists, one pharmacist, and two physicians) were conducted to gather further information for some input parameters in case there is no data. Technical validation was also performed by (1) inviting an external health economist to check the model construction; and (2) conducting trace analysis and extreme value analysis. All experts provided their written informed consent before participating in the study.

The final model included three health states: disease-free state (DFS), DFS with cardiac events (i.e., congestive heart failure), local-regional recurrence, DFS after local-regional recurrence, metastatic stage, and death (see Fig 1). We applied a cycle length of one year. Upon administration of either chemotherapy with or without trastuzumab, the patient cohort enters the model at the DFS. Each year, the patient could move between different health states with a certain probability, as indicated by the arrows in Fig 1. The model was constructed using Microsoft Excel.

## 2.4. Key assumptions

The following assumptions were made in the analysis, similar to previous studies. All assumptions were consulted with experts for validation purposes. We assumed that the benefit of trastuzumab treatment lasts five years and that patients receiving adjuvant trastuzumab therapy experience cardiotoxicity within the first year of treatment [29]. There were no breast cancer

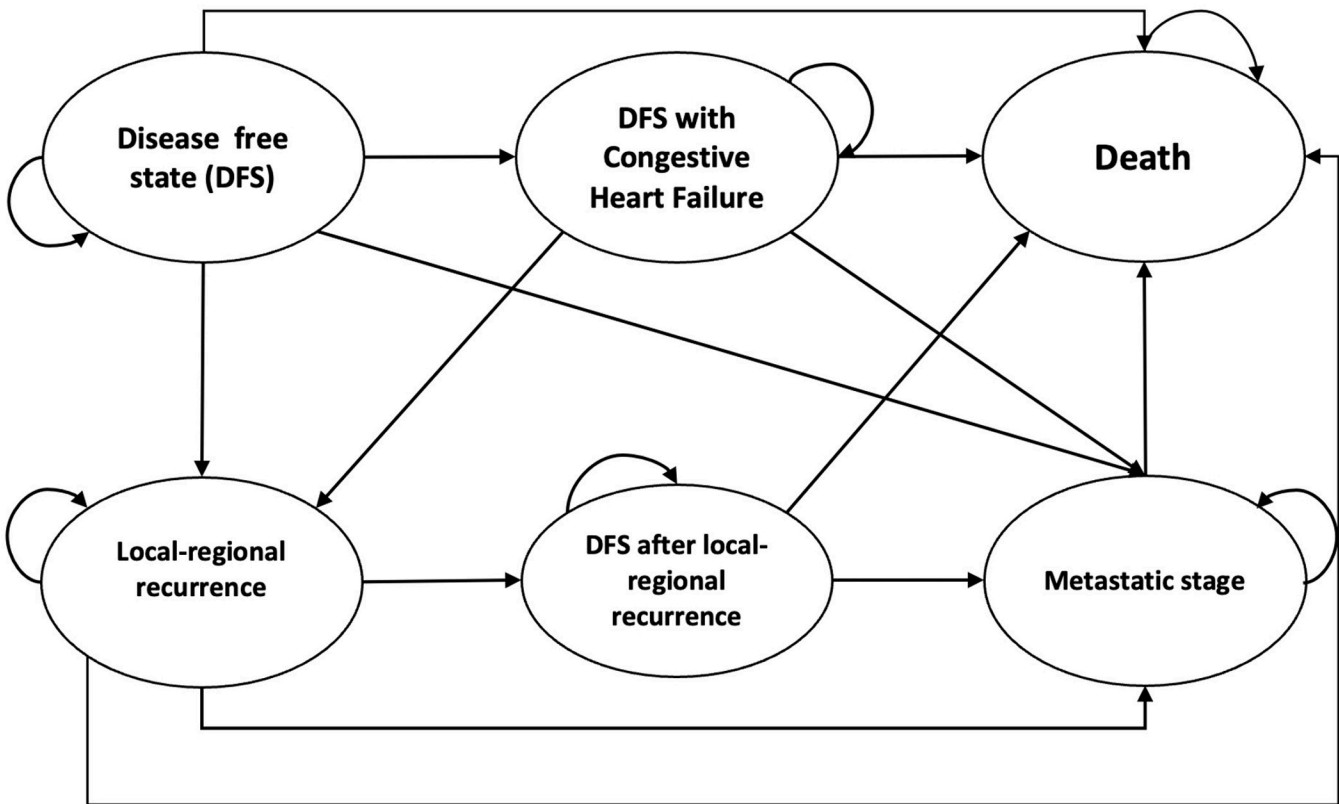

**Fig 1. Markov model.**

recurrences beyond year 20 of the model. Patients in all health states could die due to background mortality. Patients who had progressed to local-regional recurrence after receiving the treatment could move to the DFS after the local-recurrence state; patients who went to the metastatic stage remained until they died from breast cancer or other causes [29]. We assumed that death from breast cancer occurred only in patients in the metastatic stage [24]. The mortality due to CHF was supposed to be zero [29]. Patients in all health states could die due to background mortality. We assumed patients in both arms were equally likely to receive all other medical services. We did not consider the cost of trastuzumab treatment for patients in the metastatic stage because the very low percentage of patients in Vietnam will continue to receive trastuzumab.

### 2.5. Input parameters

**2.5.1. Clinical data.** The effectiveness of trastuzumab in combination with chemotherapy compared with chemotherapy alone was presented as a hazard ratio (see Table 1 for details). Effectiveness data were obtained from clinical trials N9831 and NSABP B-31 (4-year follow-up) [28,30], HERA—the Herceptin Adjuvant Trial [19,27,31], BCIRG 006—the Breast Cancer International Research Group trial 006 [29]. Transition probability was referenced from the above clinical trial and studies [26,29]. Details on clinical parameters are presented in Table 1. Transition probabilities among patients receiving chemotherapy alone were determined first, and then transition probabilities among patients receiving trastuzumab in combination with chemotherapy were adjusted based on the hazard ratios. The transition probabilities were developed separately for the first year, 2nd-5th year, and from year six onwards.

**Table 1. Input parameters.**

| Input parameters | Basecase value | Sources |
|---|---|---|
| **Clinical parameters** | | |
| *Treatment efficacy* | | |
| DFS HR in the trastuzumab arm | 0.760 | Cameron et al., 2017 [19] |
| Local-regional recurrence HR in the trastuzumab arm | 0.580 | Perez et al., 2014 [30] |
| Metastatic HR in the trastuzumab arm | 0.480 | Romond, 2005 [28] |
| **Baseline transition probabilities** | | |
| Local-regional recurrence without trastuzumab | 0.0294 | Buendia JA, 2013 [26] |
| Metastasis rate without trastuzumab | 0.0785 | Piccart-Gebhart et al., 2005 [31] |
| DFS after local-regional recurrence | 0.100 | Buendia JA, 2013 [26] |
| Increased risk of metastatic after local-regional recurrence | 3.640 | de Bock et al., 2009 [32] |
| *Cardiac event* | | |
| Trastuzumab-induced CHF | 0.198 | Slamon, 2011 [29] |
| CHF due to chemotherapy | 0.007 | Slamon, 2011 [29] |
| *Mortalities* | | |
| Death from metastasis stage | 0.295 | Piccart-Gebhart et al., 2005 [31] |
| Age-specific background mortality rates | - | WHO life table for Vietnam, 2020 [33] |
| **Utilities** | | |
| DFS | 0.832 | Based on a quality of life survey in 2014 [23] and patients survey in 2015 |
| Local regional recurrence | 0.828 | Patient survey in 2015 |
| DFS after local regional recurrence | 0.789 | Patients survey in 2015 and expert opinion |
| DFS with CHF | 0.670 | Based on systematic review [34] and expert opinion |
| Metastatic stage | 0.762 | Based on a quality of life survey in 2014 [23] and patients survey in 2015 |
| **Cost in VND (US$)** | | |
| *One-year adjuvant trastuzumab therapy* | | |
| Cost for detecting HER2 | 686,116 (30,165) | Calculation based on Circular 13/2019/TT-BYT and experts' opinion |
| Cost of trastumumab | 787,384,650 (34,618) | Drug Administration of Vietnam |
| Other direct medical costs (including para-clinical services, hospitalization, outpatient visits, consumable and other direct medical cost) | 27,040,970 (1,188) | Calculation based on [16] and expert opinion |
| Direct non-medical costs | 5,724,180 (252) | Calculation based on [16] |
| *Cost of paclitaxel treatment alone* | 32,116,418 (1,412) | Calculation based on [16] and experts' opinion |
| *One-year cost of other health states* | | |
| Cost incured by patients in DFS | 1,400,846 (62) | Calculation based on [16] |
| Cost incured by patients in Local-regional recurrence | 216,688,208 (9,526) | |
| Cost incured for patients in DFS after Local-regional recurrence | 1,400,846 (62) | |
| Cost incured for patients in Metastasis stage | 131,065,482 (5,762) | |
| Cost of incured for patients in DFS with CHF | 9,529,840 (418) | Calculation based on [16] and expert opinion |

Note: HR/harazd ratio, DFS/disease free state, DAV/Drug Administrative, CHF/Congestive heart failure.

**2.5.2. Utility data.** The utility values associated with the model health states were mainly sourced from a quality of life survey of 376 breast cancer patients enrolled from three oncology hospitals across Vietnam in 2014 [23]. Within this project, we also conducted a small survey on 56 breast cancer in-patients and 30 breast cancer out-patients in two oncology hospitals in Hanoi capital (located in the north of Vietnam) and Da Nang (located in the south of Vietnam). Both of the surveys employed the EQ-5D-5L questionnaire to describe health states. Then, based on the survey results, the utility values were calculated using the Korean scoring system [35]. This scoring system has been shown to be applicable to Vietnam [36]. Table 1 presents the utility values for different health states in the analysis. Electronic Supplement (S2 Text) also demonstrates the utility values collected in the patient survey in 2015.

**2.5.3. Costs data.** *Cost of adjuvant trastuzumab therapy*: The cost of trastuzumab treatment was calculated for one patient testing for HER2+ status until the completion of therapy after one year. Patients were assumed to have a regular cardiovascular function assessment every four months; and routine haematology and biochemical tests once per trastuzumab transfusion. It was assumed that 100% of patients have an immunohistochemical (IHC) for detecting HER2 status. Among patients taking the IHC test, 5% of the patients with IHC3 + results, 4% with IHC2+ results, and 7% with IHC1+/- results were assumed to proceed with the FISH test. Patients using trastuzumab were covered by health insurance at 60% of the trastuzumab cost, and 60% of the FISH test cost. Corresponding to the proportion of patients having social health insurance in the previous patient surveys [16,23], it was assumed that 100% of patients have social health insurance, of which 9%, 34%, and 57% were paid 100%, 5% and 80% of the hospital fees, correspondingly.

Fees for laboratory tests, general examination, and inpatient beds were collected from the current validated national health service fee schedule (Circular 13/2019/TT-BYT approved by the Vietnam Ministry of Health since 20/09/2019). Since the fees do not fully cover the total cost given that public hospitals receive government subsidies, we also included the costs of services from the government perspective (including government subsidy for labor costs, depreciation of assets, and overhead cost to perform laboratory tests, outpatient visit, and inpatient care). The costs of services from the government perspective were determined through a standard activity-based costing study, utilizing secondary data gathered from four major oncology hospitals in Vietnam in 2014. These hospitals included the Vietnam National Cancer Hospital, Hanoi Oncology Hospital, Da Nang Oncology Hospital, and Ho Chi Minh City Oncology Hospital [16]. Cost of other direct medical costs (e.g., consumables) and direct non-medical costs (transportation) were also collected from patient surveys [16,37]. The price of trastuzumab and other medicines for supportive treatment (e.g., vitamins) was updated on the website of the Drug Administration of Vietnam (DAV). Details of values and sources of all cost parameters are presented in the Electronic Supplement (S2 Table). All cost data were updated in 2020. The results of calculating the cost of trastuzumab treatment for one patient in 1 year of treatment are presented in Table 1. Detailed explanations for the costing process were provided in the Electronic Supplement (S1 Text).

*Cost of congestive heart failure*: Assuming that after the event of CHF, the patient discontinued using trastuzumab and was treated with the mean hospitalization time of two weeks and the outpatients' management duration of three months. We identified the cost items based on the American Heart Association treatment guideline (also followed by Vietnam physicians as current practice). The cost quantities were estimated based on consultation with a physician expert. The unit costs (in 2020 value) for identified cost items were collected from the most updated sources, including DAV's website and previous patient surveys. The input parameters are detailed in the Electronic Supplement (S2 Table). Estimates are presented in Table 1.

*Other costs*: For the chemotherapy alone option, the cost of paclitaxel treatment was calculated, including (1) the cost of paclitaxel, (2) the cost of testing; (3) the cost of inpatient treatment for drug infusion; (4) costs of infusions and adjuvant drugs; (5) other direct medical costs and (6) direct non-medical costs for using chemotherapy. For the treatment option of adjuvant trastuzumab therapy in combination with chemotherapy, it was assumed that all patients were treated simultaneously with trastuzumab and paclitaxel during hospitalization. Therefore, only the cost of the drug paclitaxel was considered in this case. Secondary data analysis from the patient survey was performed to collect data on the cost of other health states. Estimates are presented in Table 1.

*Expected value of information analysis*: The expected value of perfect information (EVPI) for the whole decision is assessed. As the optimal choice of intervention was highly certain, there was no value in collecting additional information, reflected in the relatively low value of perfect information. Detailed explanations for the value of information analysis were provided in the Electronic Supplement (S3 Text).

## 2.6. Method of analysis and presentation of results

Results are reported as incremental cost per Life Year (LY) gained and Quality Adjusted Life Year (QALY) gained with one-year adjuvant trastuzumab combined with chemotherapy compared to chemotherapy alone. We applied the discounting rate of 3% per year to costs and outcomes as recommended by Vietnam Health Technology assessment guideline [15]. The cost was measured in terms of Vietnam dong (VND). We converted the cost in VND to 2020 United States dollars (US$) using the exchange rate of US$ 1 = 22,745 VND. We applied the current Vietnam cost-effectiveness threshold values of 83, and 249 million VND per QALY gained (roughly US$ 3,600 and US$ 10,800) that were widely accepted in Vietnam based on the value of one time and three times Gross Domestic Product (GDP) per capita in Vietnam in 2020. In the case that trastuzumab was not cost-effective, a threshold analysis was performed to calculate the cost-effective price of trastuzumab. Scenario analyses were also performed to inform the policy-making process.

Both one-way and probabilistic sensitivity analyses (PSA) using Monte Carlo simulation replicated for 10,000 iterations were performed to handle parametric uncertainties. Clinical values varied along the 95% confidence intervals reported in the published literature. According to experts, utility and cost values were altered by 10% around the base case value or between the upper and lower frontier (See Electronic Supplement (S3 Table) for more details). We assigned beta distribution for transition probabilities and utility parameters, log-normal distribution for the relative treatment effect, and gamma distribution for cost parameters. The PSA results were illustrated as incremental cost-effectiveness planes and cost-effectiveness acceptability curves.

## 3. Results

### 3.1. Base case results

Based on the deterministic analysis, from the societal perspective, adjuvant trastuzumab therapy combined with chemotherapy compared to chemotherapy alone was estimated to incur an additional lifetime cost of 4,216,100 million VND (corresponding to 185,363,845 US$, which equals 0.053% of total GDP in 2020 in Vietnam) with an additional lifetime health gain of 15,618 LYs or 8,113 QALYs for the whole hypothetical population of 5,052 early-stage breast cancer women with HER2+ in Vietnam (Table 2). For each patient, trastuzumab, compared to chemotherapy alone, yielded an additional cost of 888,453,971VND (39,062 US$) with an additional 3.09 LYs and 1.61 QALYs, resulting in an ICER of 287,390,682 VND (12,635 US$) per LY gained, or 519,616,972 VND (22,845 US$) per QALY gained. Thus, adding one year of

**Table 2. Base case results.**

| | Trastuzumab combined with chemotherapy | Chemotherapy alone | Difference |
|---|---|---|---|
| Total lifetime costs (million VND) | 6,224,807 | 2,008,706 | 4,216,100 |
| Total lifetime costs (US$) | 273,678,047 | 88,314,202 | 185,363,845 |
| Total LYs gained | 57,530 | 41,912 | 15,618 |
| Total QALYs gained | 44,112 | 35.998 | 8,113 |
| ICER (VND per LY gained) | 287,390,682 | | |
| ICER (US$ per LY gained) | 12,635 | | |
| ICER (VND per QALY gained) | 519,616,972 | | |
| ICER (US$ per QALY gained) | 22,845 | | |

Notes: VND/Vietnam dong; US$/United State dollar; LY/Life Year; QALY/Quality-Adjusted Life Year; ICER/Incremental cost effectiveness ratio.

adjuvant trastuzumab to chemotherapy treatment for HER2+ early-stage breast cancer patients is not cost-effective in Vietnam as the ICERs of 519 million VND per QALY gained exceed the cost-effective threshold of 1-time GDP per capita and the threshold of 3-time GDP per capita by 6.3 times and 2.1 times, correspondingly.

## 3.2. Sensitivity analysis

The results of the one-way sensitivity analysis are presented as a tornado diagram (Fig 2). The ICER results were most sensitive to variations in transition probabilities from DFS with CHF to local-recurrence and metastasis state, price of trastuzumab 150mg vial and 400mg vial, the adjusted hazard ratio for recurrence with trastuzumab, the relative risk for development of metastasis with trastuzumab compared to chemotherapy alone.

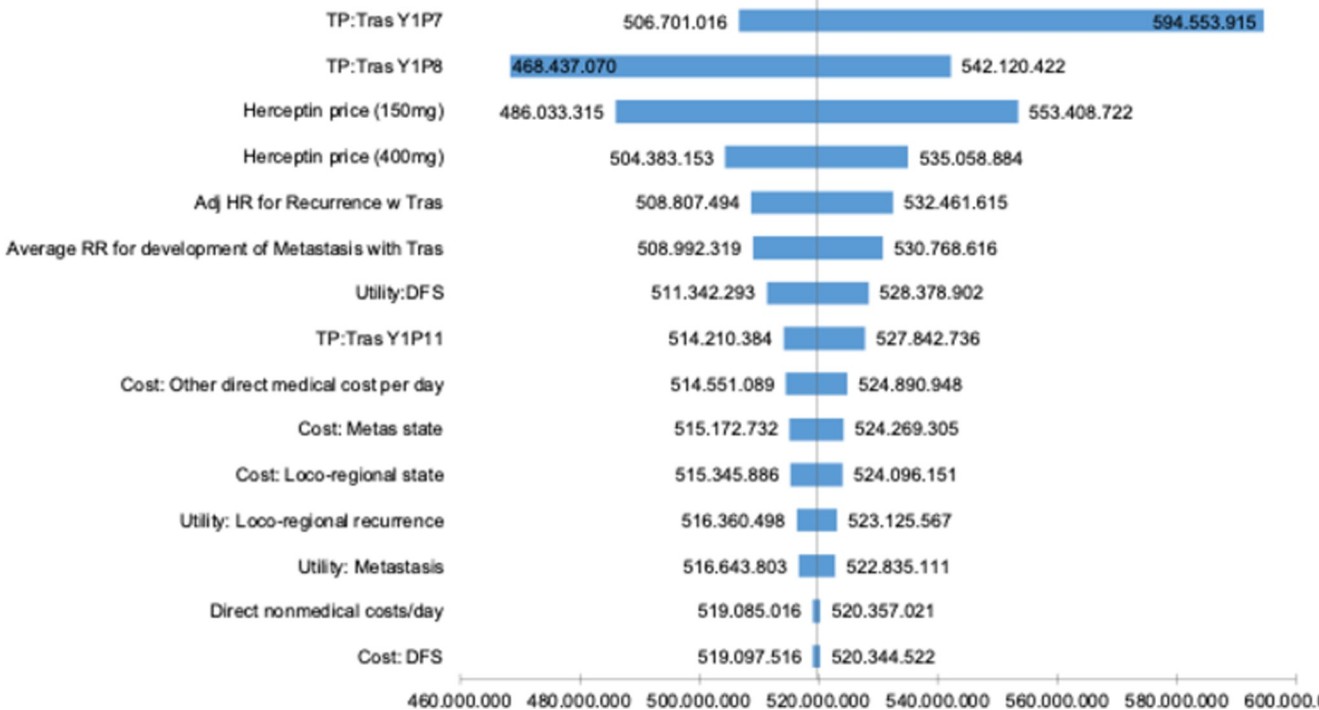

**Fig 2. Tornado diagram of one-way sensitivity analysis.**

The results of the Monte Carlo simulation with 10,000 replications were plotted in a cost-effectiveness plane (Fig 3). All of the ICER plots appear at the upper-right-hand quadrant of the plane. The single black square represents the resulting mean ICER while the single black line and double black line represent the cost-effectiveness thresholds of 1-time GDP per capita and 3-time GDP per capita, where estimates below these lines are considered cost-effective.

Fig 4 presents the cost-effectiveness acceptability curve. It demonstrates that at the cost-effectiveness thresholds of both one time and three times per capita (i.e., 83 million VND and 249 million VND) per QALY gained, the probability of cost-effectiveness of adjuvant trastuzumab therapy is 0%, while that for the chemotherapy alone regimen is 100%. The current cost-effectiveness threshold, i.e., the current GDP per capita in Vietnam, must increase by about six times (i.e., 500 million VND per QALY gained) for adjuvant trastuzumab to reach a 50% probability of cost-effectiveness.

### 3.3. Scenario analysis

In Vietnam, trastuzumab is used under the only brand name Herceptin, primarily available as an intravenous (IV) formulation in 150mg and 440mg vials. The current price of Herceptin 150mg and 440mg vials are 15,550,710 VND (684 US$) and 45,596,775 VND (2,005 US$). Because only some eligible patients received Herceptin simultaneously (and not all hospitals have a centralized drug dispensing system), the Herceptin 150mg and 440mg vials could not be shared with many patients. For example, with an average weight of 50kg, the medication needed was 400mg (8mg/kg body weight). If there was only one patient, one vial of Herceptin 440mg must be used with the excess 40mg discarded. Meanwhile, this 40mg could be shared with others if treating multiple patients simultaneously.

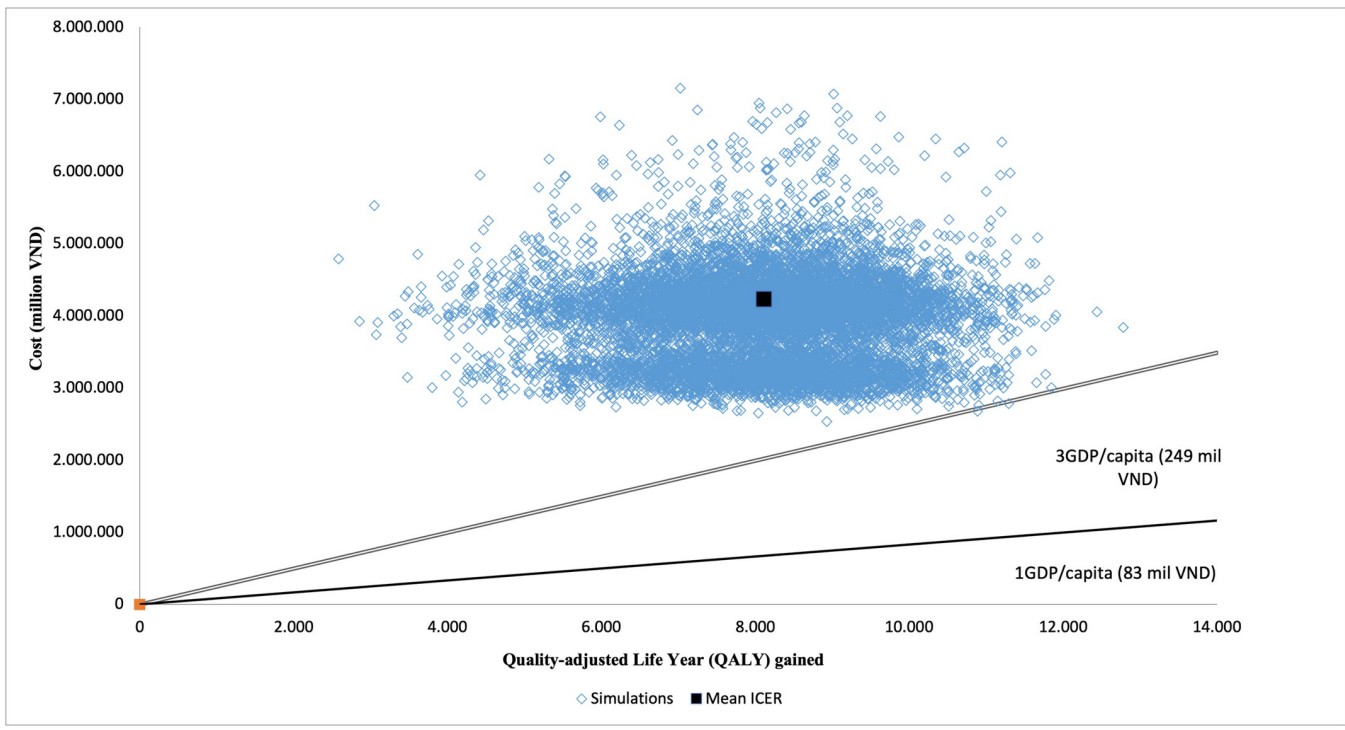

**Fig 3. Scatter plot of probabilistic results.**

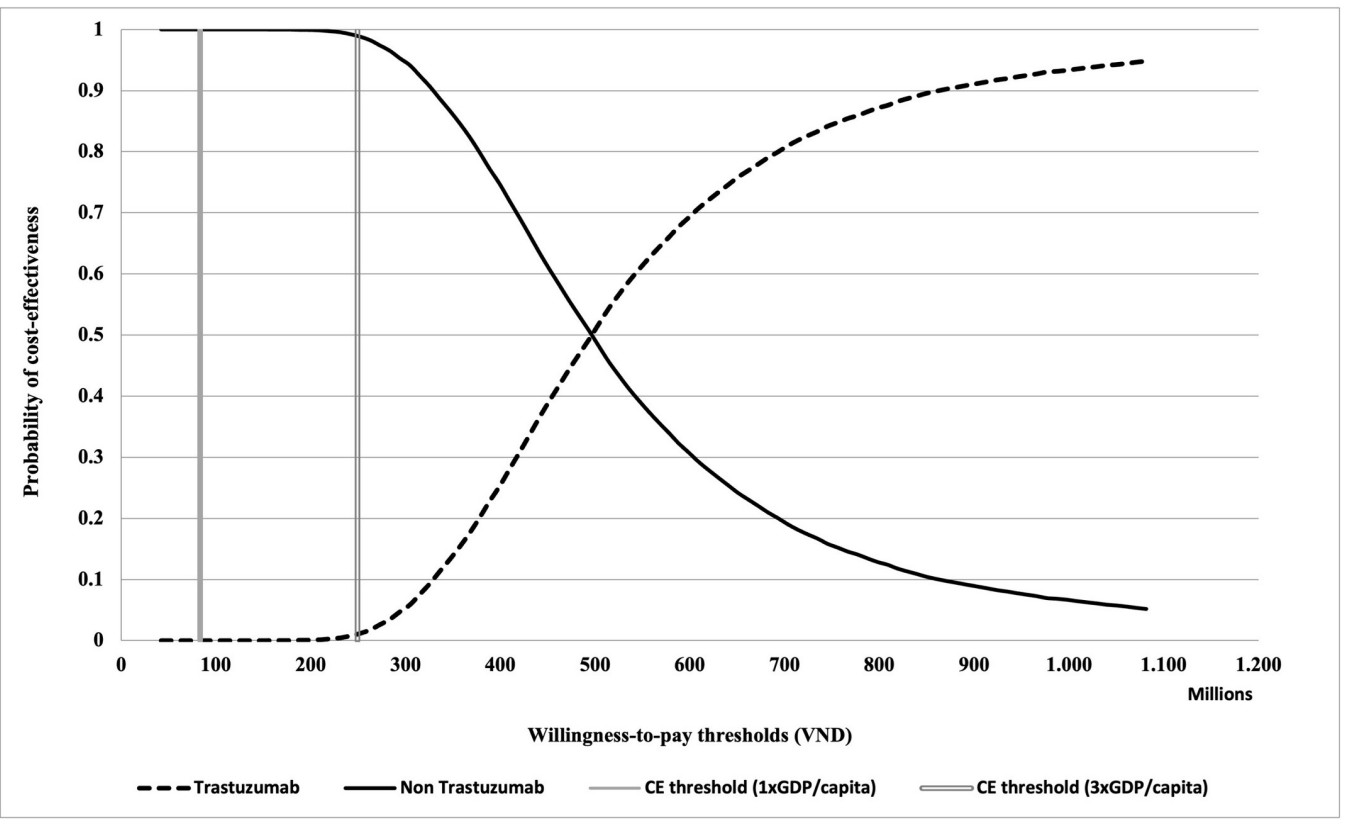

**Fig 4. Cost-effectiveness acceptability curve.**

Therefore, based on recommendations from the expert panel (i.e., from the oncologists and patients' representatives), we performed the scenario analysis when the 150mg and 440mg vials were shared to avoid waste. Table 3 summarizes the results of the scenario analysis. If there are enough patients at the same time to share the 150mg and 440mg vials, the mean cost of one-year trastuzumab (drug only) was reduced to 570,022,437 VND (25,061 US$) compared to the base case of 787,384,650 VND (34,618 US$). Consequently, ICER per QALY gained from one-year adjuvant trastuzumab therapy compared to chemotherapy alone was reduced to 384,278,974 VND (16,895 US$), which is still not cost-effective since the ICER exceeded the current threshold of 1-time GDP per capita and the threshold of 3-time GDP per capita. The probability of adjuvant trastuzumab therapy cost-effectiveness at the threshold level of 1-time GDP per capita and 3-time GDP per capita was 0% and 2%, respectively.

### 3.4. Threshold analysis

Table 3 also demonstrates the results for threshold analysis in both base-case (i.e., the Herceptin 150mg and 440mg vials could not be shared among patients) and the scenario (i.e., the Herceptin 150mg and 440mg vials could not be shared among patients). For the option of using one-year adjuvant trastuzumab therapy to be cost-effective at the threshold level of 1-time GDP per capita, the price of Herceptin 150mg and 440mg vials must be less than 1,689,167 VND (74 US$) and 5,067,500 VND (223 US$) in the base case and 2,349,874 VND (103 US$) and 6,892,964 VND (303 US$) in the shared options. Similarly, Table 3 presents the maximum

**Table 3. Results of the scenario and threshold analysis.**

| | Not share the drug vial (Base case) | | Shared the drug vial (Scenario) | |
|---|---|---|---|---|
| | VND | US$ | VND | US$ |
| Total cost of one-year adjuvant trastuzumab therapy (per person) | 820,835,918 | 36,088 | 603,473,704 | 26,532 |
| Cost of trastuzumab (drug only) | 787,384,650 | 34,618 | 570,022,437 | 25,061 |
| ICER per QALY gained | 519,616,972 | 22,845 | 384,278,974 | 16,895 |
| Probability of trastuzumab becomes cost-effectiveness | | | | |
| At threshold of 1-time GDP per capita | 0% | | 0% | |
| At threshold of 3-time GDP per capita | 0% | | 2% | |
| *Threshold analysis* | | | | |
| When ICER is set at 1-time GDP per capita | | | | |
| Price of trastuzumab 440mg vial | 5,067,500 | 223 | 6,892,964 | 303 |
| Price of trastuzumab 150mg vial | 1,689,167 | 74 | 2,349,874 | 103 |
| When ICER is set at 3-time GDP per capita | | | | |
| Price of trastuzumab 440mg vial | 20,750,295 | 912 | 28,225,165 | 1,241 |
| Price of trastuzumab 150mg vial | 6,916,765 | 304 | 9,622,215 | 423 |

Notes: Current price of Herceptin 440mg = 45,596,775 VND (2,005 US$); Current price of Herceptin 150mg = 15,550,710 VND (684 US$); 1-time GDP percapita threshold = 83 million VND (3,600 US$), 3-time GDP per capita threshold = 249 million VND (10,800 US$); QALY/Quality-adjusted life year; ICER/Incremental cost-effectiveness ratio, GDP/Gross Domestic Product.

price of Herceptin 150mg and 440mg vials at which one-year trastuzumab therapy is cost-effective at the threshold level of 3-time GDP per capita.

## 4. Discussion

This study is the first cost-effectiveness analysis of one-year adjuvant trastuzumab therapy for treating early-stage breast cancer in Vietnam. The development and validation of the Markov model were carried out following the standard steps recommended by international guidelines [38,39]. The model structure, assumptions, and data sources for input parameters were carefully discussed with experts to present all the best available evidence appropriated to the Vietnam context. Thus, the study provides significant and reliable evidence for policymakers in deciding to what extent to include trastuzumab, an effective but very high-cost drug, in the SHI benefit package.

Our findings from deterministic, PSA, and scenario analysis confirm that one-year adjuvant trastuzumab therapy is not cost-effective compared to chemotherapy alone for early-stage breast cancer with HER2+ at its current price of trastuzumab in Vietnam in 2020 from the societal perspective. Adding one year of adjuvant trastuzumab to chemotherapy treatment for HER2+ early-stage breast cancer patients incurred 22,845 US$ per QALY gained, which exceeded the cost-effective threshold of 1-time GDP per capita (i.e., 3,600 US$) by 6.3 times. In a more efficient way of administering trastuzumab for patients (i.e., sharing the Herceptin vials among concurrent hospitalized patients), the ICER was reduced to 16,895 US$, which still exceeded the threshold by 4.6 times. The reported ICERs in our study are even lower than those reported in similar studies. Indeed, a newly published systematic review [14] found 22 cost-effectiveness analysis studies of adjuvant trastuzumab therapy for early breast cancer published from 2007 to 2017. Of 22 studies, only six were conducted in upper-middle-income countries (e.g., China, Colombia, Iran, Argentina, Bolivia, and Thailand), similar to the Vietnam context. The systematic review concluded that ICERs from high-income countries were

within their cost-effectiveness thresholds and ranged from 6,018 to 78,929 US$ (in 2017) per QALY gained; meanwhile, ICERs from upper-middle-income countries ranging from 3,526 US$ up to 174,901 US$ (in 2017), which mostly exceeded their corresponding thresholds (i.e., ranging from 5,148 US$ to 34,092 US$) [14].

It is worth noting that all of the above thresholds used in the upper-middle income countries are much higher than our threshold of 1-time GDP per capita. Moreover, most of the included studies in the review were performed from a healthcare provider perspective, which would significantly undermine the true benefit of the intervention in society [13,14]. Compared to a similar cost-effectiveness analysis employing the societal perspective conducted in a similar lower-middle-income country, i.e., Philipin, the ICER reported in our study seems to be still higher than their estimation, which is about 9,800 US$ (in 2020 US$) [40]. It is difficult to compare the ICERs among countries due to the high heterogeneity in the economic background, healthcare systems, breast cancer burden, and study methodology. Our study enriches the current literature on the cost-effectiveness of adjuvant trastuzumab therapy in a low-middle income context. Our study confirms that one-year adjuvant trastuzumab therapy is not cost-effective due to its high cost in a resources-constrain context like Vietnam and other low-middle-income countries.

One-year adjuvant trastuzumab therapy would be cost-effective at the threshold of 3-time GDP per capita if the cost of Herceptin 150mg and 450mg vials were reduced by 56% and 54%, correspondingly. Even when sharing the Herceptin vials among patients, trastuzumab use would be cost-effective (at the threshold of 3-time GDP per capita) only if the price of Herceptin vials were reduced by 38%. The ability to share the Herceptin vials needed to be carefully considered because there was not a large enough number of patients to be treated simultaneously and it required a centralized drug dispensing system that was not available in all oncology hospitals.

The result from threshold analysis in our study could support the policymakers in price negotiation with the pharmaceutical company to make this effective drug more affordable. The policymakers also need to consider the availability of biosimilars to trastuzumab originator (i.e., Herceptin) with no statistical difference in the overall response rate and complete pathological response [41]. Meanwhile, the availability of trastuzumab biosimilars could reduce trastuzumab price, resulting in better cost-effectiveness of using trastuzumab in Vietnam.

A shorter treatment duration (i.e., six months) of trastuzumab was cost-effective compared to the standard one-year treatment of adjuvant trastuzumab [42]. In this cost-effectiveness analysis, the efficacy of a six-month treatment compared to a one-year trastuzumab treatment was based on one single RCT (i.e., PHARE trial) [43]. Recently, another RCT (i.e., PERSEPH-ONE trial) also confirmed the similar efficacy of a six-month trastuzumab treatment compared to a one-year treatment [44]. As discussed in a previous systematic review [13], once findings from other ongoing trials on the efficacy of a shorter duration of treatment are available, the use of a shorter treatment duration could achieve greater cost-effectiveness of trastuzumab therapy. Shortly, a further cost-effectiveness analysis on a shorter treatment duration of trastuzumab should be conducted in Vietnam to see if trastuzumab cost could be reduced meanwhile its effectiveness maintains. Suppose a shorter treatment duration is as efficacious as the year-long course while achieving greater cost-effectiveness. In that case, policymakers should consider acknowledging not only the optimal treatment duration in terms of efficacy but also the optimal duration in terms of cost-effectiveness.

There are some limitations in our study. Firstly, this study used results from RCTs conducted in Western countries (e.g., the effectiveness of trastuzumab in breast cancer patients with HER2+ and the probability of developing local regional or distant metastasis), which may not reflect the Vietnam context. Secondly, due to a lack of updated data on utilities, we must

employ quality of life results from a breast cancer patient survey since 2014, a small size patient survey since 2015 (56 in-patients and 30 out-patients), and borrow the value for DFS with CHF and metastasis state from other countries. The utility data from different sources may not accurately represent the utility of the specific population in Vietnam. For instance, it was noticed that the utility value for the disease-free state following loco-regional recurrence (0.789) was lower than during the recurrence (0.828). This discrepancy is likely attributed to variations in research methodologies, limited sample sizes, participant demographics, and the context of the studies. However, despite this current limitation on lack of Vietnamese clinical data and some utility values, the current ICERs are so far above the current threshold, i.e., 100% of the ICERs are higher than the threshold of 1-time GDP per capita (US$ 3,600) and 99,98% of the ICERs are higher than the threshold of 3-time GDP per capita (US$ 10,800) in sensitivity analysis, making the conclusion is likely to be robust.

Thirdly, due to a lack of cost data, we must employ a bottom-up approach to calculate the cost of trastuzumab and paclitaxel treatment by gathering data from various sources. Some cost parameters, such as services unit cost and direct non-medical cost, were obtained from data sources dated back to 2014, which may have significantly increased by 2023. Despite the fact that we presented all data sources and consulted with experts to ensure the reliability of data used, updated all possible data sources, conducted inflation adjustments for 2014 cost data and performed comprehensive sensitivity analysis, the task of cost calculation remains labor-intensive and in need of improvement in future research. Researchers need to further their study on costing to provide updated input to facilitate the implementation of related health technology assessment shortly. Lastly, because of the unavailability of data regarding the indirect costs, such as productivity loss resulting from work absenteeism, short-term and long-term disability, within this specific patient group in Vietnam, we opted not to incorporate this cost component in our calculations. This designates the study's perspective as a restricted societal perspective, rather than a true societal perspective. Omitting the estimation of productivity loss was a deliberate choice, as such calculations can entail subjective judgments and lack standardized metrics for accuracy. The study results, even without factoring in indirect costs, may thus be more comprehensible and palatable within the context of funding decision-making. By omitting the estimation of indirect costs, the study results may be more readily accepted and considered within the context of resource allocation by policymakers and stakeholders.

## 5. Conclusion

In Vietnam, the option of using trastuzumab in combination with chemotherapy for early breast cancer patients with HER2+ is not cost-effective compared to chemotherapy alone, with an ICER value of 519,616,972 VND (US$ 22,845) per QALY gained, which is six times higher than the current threshold of 1-time GDP per capita (i.e., US$ 3,600). The policymakers should consider the options to reduce the cost of trastuzumab, such as using trastuzumab biosimilars, price negotiation options, and optimizing the use of Herceptin vials among concurrent hospitalized breast cancer patients. Only by reducing the price of the drug by at least 38% (in the case of Herceptin vials could be most effectively shared among concurrent patients with the cost-effectiveness threshold of 3-time GDP per capita), the use of trastuzumab may be considered as a cost-effective option compared to chemotherapy alone.

## Supporting information

**S1 Checklist.**
(DOCX)

**S1 Table. CHEERS 2022 checklist.**
(DOCX)

**S2 Table. Input parameters for cost calculation.**
(DOCX)

**S3 Table. List of variables used in one-way sensitivity analysis.**
(DOCX)

**S1 Text. Cost analysis.**
(DOCX)

**S2 Text. Utility values.**
(DOCX)

**S3 Text. Value of information analysis.**
(DOCX)

## Author Contributions

**Conceptualization:** Anh Quynh Nguyen, Oanh Thi Mai Tran, Phuong Khanh Nguyen, Ha Thu Nguyen.

**Data curation:** Ha Thu Nguyen.

**Formal analysis:** Anh Quynh Nguyen, Ha Thu Nguyen.

**Funding acquisition:** Oanh Thi Mai Tran.

**Methodology:** Anh Quynh Nguyen, Phuong Khanh Nguyen, Ha Thu Nguyen.

**Project administration:** Oanh Thi Mai Tran, Phuong Khanh Nguyen.

**Resources:** Oanh Thi Mai Tran.

**Software:** Anh Quynh Nguyen.

**Supervision:** Anh Quynh Nguyen, Oanh Thi Mai Tran, Phuong Khanh Nguyen.

**Validation:** Anh Quynh Nguyen, Ha Thu Nguyen.

**Visualization:** Ha Thu Nguyen.

**Writing – original draft:** Anh Quynh Nguyen, Oanh Thi Mai Tran, Phuong Khanh Nguyen, Ha Thu Nguyen.

**Writing – review & editing:** Anh Quynh Nguyen, Oanh Thi Mai Tran, Phuong Khanh Nguyen, Ha Thu Nguyen.

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
