## [Decision Letter · Decision Letter 0]

5 Jul 2023

PONE-D-23-09955Cost-effectiveness of one-year adjuvant Trastuzumab therapy in treatment for early-stage breast cancer patients with HER2+ in VietnamPLOS ONE

Dear Dr. Nguyen,

Thank you for submitting your manuscript to PLOS ONE. After careful consideration, we feel that it has merit but does not fully meet PLOS ONE’s publication criteria as it currently stands. Therefore, we invite you to submit a revised version of the manuscript that addresses the points raised during the review process.

We look forward to receiving your revised manuscript.

Kind regards,

George Mugambage Ruhago, PhD

Academic Editor

PLOS ONE

Journal Requirements:

“The development of the Markov model employed in this study was initiated from the Vietnam National Health Technology Assessment project period 2014-2018. The project was coordinated by the Vietnam Health Strategy and Policy Institute (HSPI), a unit under the Vietnam Ministry of Health responsible for providing evidence and consulting for the health policy-making process in Vietnam. The topic was selected based on the topic selection process for the Vietnam HTA project conducted by the HSPI.”

5. Please ensure that you refer to Figure 2 in your text as, if accepted, production will need this reference to link the reader to the figure.

6. Please include a separate caption for each figure in your manuscript

Reviewers' comments:

Reviewer's Responses to Questions

**Comments to the Author**

1. Is the manuscript technically sound, and do the data support the conclusions?

Reviewer #1: Partly

Reviewer #2: Yes

2. Has the statistical analysis been performed appropriately and rigorously? 

Reviewer #1: Yes

Reviewer #2: Yes

3. Have the authors made all data underlying the findings in their manuscript fully available?

Reviewer #1: Yes

Reviewer #2: Yes

4. Is the manuscript presented in an intelligible fashion and written in standard English?

Reviewer #1: Yes

Reviewer #2: Yes

5. Review Comments to the Author

Reviewer #1: Title should change to indicate that the chemotherapy used that is paclitaxel unless a scenario analysis with other chemotherapy is conducted

Cost - It is not clear why the authors chose to use 2014 cost information. Primary cost data could left the results more as a number of resource use elements could have changed between 2014 and 2023

The two key resources cited could not be retrieved to verify the cost methodology employed since societal perspective is said to be used. Are these published in peer review journals? Authors should consider summarizing the methods used for cost estimation in the methods section

Cost perspective is said to be societal, page 7 line 21 communicates a different perspective

High-quality evidence from multiple model input sources is not always available - in this case the use of 2014 cost data and decisions based on these model recommendations are subject to costly uncertainty. Authors should consider estimating the cost of existing uncertainty in terms of value of added information to check if it is potentially cost effective to seek for more evidence

Reviewer #2: PONE-D-23-09955

Cost-effectiveness analysis of Trastuzumab in treatment for early-stage breast cancer patients with HER2+ in Vietnam

Thank you for the opportunity to review this paper describing a cost-utility analysis of trastuzumab in a specific breast cancer patient group, in Vietnam. This appears to be a well-conducted study, using advisory groups to review the model design and input parameters. The study adds to the literature on the economics of trastuzumab, for which funding decisions have attracted debate in most jurisdictions.

My main concern with this work is the discussion of limitations, – and they are important limitations: lack of Vietnamese clinical data, and (some) utility values. The authors identify the limitations clearly in the Discussion, but very briefly by simply stating them. It might be helpful to readers if the authors could discuss the possible effect of these limitations on their findings; for example, is there an expected direction of bias, so do we expect the model to under-or over-estimate the ICER? Or are the ICERs so far above the threshold that the conclusion is likely to be robust regardless?

I also found the paper a little unclear on the question of perspective. The authors state that they take a societal perspective. However, there is no discussion of elements of a societal perspective such as productivity – the only component beyond the payer’s perspective appears to be inclusion of out-of-pocket payments. Further, it was not clear to me as a reader, where these out-of-pocket payments arose; there is discussion of different percentages of coverage (eg 60% of trastuzumab costs, then 100%, 5%, 80% of other fees) but without a good knowledge of the Vietnamese healthcare system and its funding, I wasn’t clear here who was paying for what. Table 1 then describes costs for each health state, that were incurred ‘by’ or ‘for’ the patient; again I was unclear how the cost-sharing works in this context. Perhaps a brief description in the Introduction, on reimbursement and coverage in the Vietnamese system, would be helpful for the international readership of PLoS-ONE.

I had two more minor observations related to the information in Table 1:

Utilities: I was surprised that the utility of the disease-free state after loco-regional recurrence was lower than during the recurrence. This is probably due to the need to use multiple sources. Could the authors make some explanation of this, in the text?

Cost estimates: for transparency and generalisability, I would like to understand what has been included in each of the cost groups (eg ‘direct medical costs’ ‘direct non-medical costs’ and the state-specific costs. The components are listed in the supporting information, but I couldn’t see how they related to the Table 1 numbers. Perhaps a simple flowchart would help?

Some minor observations and typos:

Abstract

Check for consistency of using ‘.’ or ‘,’ separators in numbers – example in the abstract (12.635 US$, should be 12,635?) and also elsewhere in the paper.

Sensitivity analysis is described as two-way, but in the Methods and Results it is descried as one-way - and the results look like a one way analysis

Introduction

I suggest checking the journal’s convention on whether to use capital letters for drug names (eg trastuzumab, paclitaxel) compared to brand names (eg Herceptin). Also check consistency of spelling of paclitaxel.

Methods

P4 line 25: reflects rather than reflexes?

Table 1: cardiac events rather than cardiact events; Hazard ratio rather than Harazd; DAV definition different from the one in the text

P5 line 6 ‘experts were given written informed consent’ – I wasn’t sure what this meant – did the experts give their consent to participate, or did someone consent to allow them to participate?

P7 lines 12 -13: I found the sentence about IHC unclear, as I’m not expert in pathology, so wasn’t sure what these groups were.

Reference year: the text states that all costs were updated in 2020 (eg P7 lines 28). Were costs collected in 2014 inflated to 2020 values?

Results

P9 line 5: vial rather than viral

Discussion

P13 line 7: findings rather than finding

P13 line 12: treatment rather than treat men

6. PLOS authors have the option to publish the peer review history of their article (what does this mean?). If published, this will include your full peer review and any attached files.

Reviewer #1: No

Reviewer #2: No

---

## [Author Response · Author response to Decision Letter 0]

14 Nov 2023

Responses to Reviewers' Comments:

Thank you very much for all of the comments. We really appreciate your comments and the comments really help us to significantly improve our manuscript. Once again, we would like to express our sincere gratitude to all of your support. 

Reviewer #1: Title should change to indicate that the chemotherapy used that is paclitaxel unless a scenario analysis with other chemotherapy is conducted

Cost - It is not clear why the authors chose to use 2014 cost information. Primary cost data could left the results more as a number of resource use elements could have changed between 2014 and 2023

*Thank you very much for your comments. We would like to explain that due to a lack of data, we must use 2014 cost information for cost from the government (Please also refer to cost element (1.1) in the subsequent response). Besides, for some of the cost elements (e.g., one year cost of disease-free state), we must collect from the 2014 patient survey (and costs were inflated to 2020 data) since there are no updated patient-level surveys on the same issues. We understand the issues that the reviewer raised and included this issue in the discussion (please see page 14, line 31-36). 

The two key resources cited could not be retrieved to verify the cost methodology employed since societal perspective is said to be used. Are these published in peer review journals? Authors should consider summarizing the methods used for cost estimation in the methods section

*Thank you very much for your comments. Quality of cost data is also our big concern. However, due to a complicated public hospital financing system in Vietnam, our unclear writing might be confusing. Thus, per the reviewer’s comments relating to cost data, we added some explanation under part 2.5.3 and summarized the cost estimation process in detail in the Electronic Supplement (Text S1: Cost calculation). 

In general, the following formula could be used to demonstrate how cost for a cost element could be estimated:

(1) Cost from societal perspective = (1.1) Cost from the government (due to government subsidizing) + (1.2) Cost from payers (social health insurance) + (1.3) Cost from patients (co-payment, out-of-pocket payment for direct medical cost as well as direct non-medical cost)

Thus, 2014 cost information (Ref #16) is the only available data source for (1.1). Ref #16 is a report written in Vietnamese and submitted to the Ministry of Health after a peer-review process (it can be retrieved by the link: https://library.huph.edu.vn/, however, it is only available in Vietnamese).

Cost perspective is said to be societal, page 7 line 21 communicates a different perspective

*Yes, we want to confirm we applied a societal perspective. Page 7 line 21 mentions the cost from the government (government subsidy), i.e., the cost element 1.1 as explained above. We also clarified the text under part 2.5.3 (page 8, line 13-19) to make it clearer. 

High-quality evidence from multiple model input sources is not always available - in this case the use of 2014 cost data and decisions based on these model recommendations are subject to costly uncertainty. Authors should consider estimating the cost of existing uncertainty in terms of value of added information to check if it is potentially cost effective to seek for more evidence

*Thank you for your suggestion, we performed the value of information analysis and presented the result in P8 lines 39-40. 

Reviewer #2: PONE-D-23-09955

Cost-effectiveness analysis of Trastuzumab in treatment for early-stage breast cancer patients with HER2+ in Vietnam

Thank you for the opportunity to review this paper describing a cost-utility analysis of trastuzumab in a specific breast cancer patient group, in Vietnam. This appears to be a well-conducted study, using advisory groups to review the model design and input parameters. The study adds to the literature on the economics of trastuzumab, for which funding decisions have attracted debate in most jurisdictions.

*Thank you very much for your comments. We really appreciate your feedback since conducting a cost-effectiveness study in Vietnam poses unique challenges, primarily stemming from data limitations and the method's relatively low recognition. Given the absence of robust cost-effectiveness evidence, trastuzumab continues to be covered by social health insurance without price negotiation. Therefore, we aspire to make a meaningful contribution to the existing literature, particularly by offering compelling evidence from Vietnam to inform strategic funding decisions. Again, we would like to show our deeply thanks since we also learn and improve our manuscrip alot based on your comments. 

My main concern with this work is the discussion of limitations, – and they are important limitations: lack of Vietnamese clinical data, and (some) utility values. The authors identify the limitations clearly in the Discussion, but very briefly by simply stating them. It might be helpful to readers if the authors could discuss the possible effect of these limitations on their findings; for example, is there an expected direction of bias, so do we expect the model to under-or over-estimate the ICER? Or are the ICERs so far above the threshold that the conclusion is likely to be robust regardless?

*Thank you for your invaluable suggestion. We added a paragraph to discuss the possible effect of the limitations on data sources (Please see Page 14, line 25-29). The ICERs are so far above the current threshold, thus the conclusion is likely to be robust in both cases. 

I also found the paper a little unclear on the question of perspective. The authors state that they take a societal perspective. However, there is no discussion of elements of a societal perspective such as productivity – the only component beyond the payer’s perspective appears to be the inclusion of out-of-pocket payments. 

*Thank you very much for your comment. Actually, we did not include indirect cost (i.e., productivity loss due to work absence, short-term and long-term disability) in the study because of the two following reasons: (1) no reliable data source on the indirect cost of this patient group in Vietnam; (2) since estimating productivity loss may involve subjective judgments (as well as lack of standardized metrics), Vietnam policymakers and stakeholders prioritize the inclusion of direct cost – both medical and non-medical cost. 

In the process of conducting this study, with the consultation of national experts and HiTAP (Thailand) as well as based on the current practice of other authors in defining “societal perspective”, we identified our study’s perspective as societal. However, reading your comment, we realized that the perspective we adopted is not a “true societal perspective”, but a “limited/restricted societal perspective” (as ISPOR recommendation in https://doi.org/10.1111/j.1524-4733.2009.00660.x). We provided some explanations on the issue in the discussion (Please see page 14, lines 37-42).

Further, it was not clear to me as a reader, where these out-of-pocket payments arose; there is discussion of different percentages of coverage (eg 60% of trastuzumab costs, then 100%, 5%, 80% of other fees) but without a good knowledge of the Vietnamese healthcare system and its funding, I wasn’t clear here who was paying for what. Table 1 then describes costs for each health state, that were incurred ‘by’ or ‘for’ the patient; again I was unclear how the cost-sharing works in this context. Perhaps a brief description in the Introduction, on reimbursement and coverage in the Vietnamese system, would be helpful for the international readership of PLoS-ONE.

*Thank you for your comments, we added a brief description in the Introduction of Vietnam health financing system and Social Health Insurance (SHI) scheme (reimbursement and coverage) (Please see page 3, lines 21-33). In general, almost all cancer patients have social health insurance (the use of private health insurance in Vietnam is rare). In terms of the SHI drug benefit package, the normal drugs can be reimbursed 100%. For expensive drugs in the benefits package, such as trastuzumab, SHI covers 60% of the drug cost, patients must co-pay the rest 40%. In terms of the benefits package for other services (such as hospitalization, surgery, and other procedures), SHI covers 80% of the cost and patients must co-pay the rest 20%. For poor or near-poor patients, SHI covers 95%-100% of the cost. 

Insured patients are not only responsible for their co-payment but also pay extra billing: (1) they pay for their “self-requested” services (the services/drugs are not included in the benefit packages); (2) they must pay for the services due to the conditions for 100% reimbursement are not fulfilled (for example, patients by-pass the lower level hospitals). Besides, patients and their relatives also pay their OOP for: (1) travelling, accomodations, foods; (2) for hiring a caregivers to take care of the patients; (3) paying for traditional medicines, or (4) paying “under-table” money, etc,. 

I had two more minor observations related to the information in Table 1:

Utilities: I was surprised that the utility of the disease-free state after loco-regional recurrence was lower than during the recurrence. This is probably due to the need to use multiple sources. Could the authors make some explanation of this, in the text?

*Thank you very much for your invaluable feedback. We also realized that it is unusual that the utility of the disease-free state after loco-regional recurrence was lower than during the recurrence. You are right that it might be due to the multiple sources we used for utility value. The quality of life survey in Vietnam (Ref#23) did not contain information to identify the DFS after loco-regional recurrence, thus we must estimate the utility for recurrence state based on a small out-patient survey (n=30, outpatients returning to the oncology hospitals for regular health visits after their loco-regional recurrence). We added some explanation of this in the discussion (Please see Page 13, lines 21-25) as your suggestion. 

Cost estimates: for transparency and generalisability, I would like to understand what has been included in each of the cost groups (eg ‘direct medical costs’ ‘direct non-medical costs’ and the state-specific costs. The components are listed in the supporting information, but I couldn’t see how they related to the Table 1 numbers. Perhaps a simple flowchart would help?

*Thank you for your suggestion, we try to provide a section to clarify our cost estimation process (Please see Electronic Supplement – Text S1)

Some minor observations and typos:

Abstract

Check for consistency of using ‘.’ or ‘,’ separators in numbers – example in the abstract (12.635 US$, should be 12,635?) and also elsewhere in the paper.

*Thank you for your comments, we revised all numbers to use “,” and “.” correctly for numbers.

Sensitivity analysis is described as two-way, but in the Methods and Results it is descried as one-way - and the results look like a one way analysis

*Thank you for your comments, it is one-way analysis and we revised the manuscript entirely for this point. 

Introduction

I suggest checking the journal’s convention on whether to use capital letters for drug names (eg trastuzumab, paclitaxel) compared to brand names (eg Herceptin). Also check consistency of spelling of paclitaxel.

*Thank you very much for raising this issue. We checked and revised the manuscript entirely, i.e., use trastuzumab and paclitaxel instead of Trastuzumab and Paclitaxel

Methods

P4 line 25: reflects rather than reflexes?

*Thank you very much for your careful readings. We corrected the mistake accordingly.

Table 1: cardiac events rather than cardiact events; Hazard ratio rather than Harazd; DAV definition different from the one in the text

*Thank you very much for your careful readings. We corrected the mistake accordingly.

P5 line 6 ‘experts were given written informed consent’ – I wasn’t sure what this meant – did the experts give their consent to participate, or did someone consent to allow them to participate?

*Thank you very much for your careful readings. We revised the sentence as follows:

All experts provided their written informed consent before participating in the study.

P7 lines 12 -13: I found the sentence about IHC unclear, as I’m not expert in pathology, so wasn’t sure what these groups were.

*Thank you for your comments. We realized the sentence confused readers. In fact, it is one of the assumptions relating to patients taking the FISH test. Not all patients take FISH because of its high cost. Thus, based on expert opinion, only 5% of patients with IHC3+ results, 4% with IHC2+ results, and 7% with IHC1+/- results obtained the FISH test. We revised the sentence in line 12-13 as follow:

“Among patients taking the IHC test, 5% of the patients with IHC3+ results, 4% with IHC2+ results, and 7% with IHC1+/- results were assumed to proceed with the FISH test.”

Reference year: the text states that all costs were updated in 2020 (eg P7 lines 28). Were costs collected in 2014 inflated to 2020 values?

*Thank you for your comments, in the sentence “All costs were updated in 2020”, we want to mention that some costs collected in 2014 were inflated to 2020 due to no 2020 data source available. However, drug prices and fees for laboratory tests and other hospitalization services were updated using 2020 data sources. We revised line P7 18-28 to clarify the meaning. 

Results

P9 line 5: vial rather than viral

*Thank you very much for your careful readings and comments. We corrected the mistake accordingly.

Discussion

P13 line 7: findings rather than finding

P13 line 12: treatment rather than treat men

*Thank you very much for your careful readings and comments. We corrected the mistake accordingly.

---

## [Decision Letter · Decision Letter 1]

27 Feb 2024

Cost-effectiveness of one-year adjuvant Trastuzumab therapy in treatment for early-stage breast cancer patients with HER2+ in Vietnam

PONE-D-23-09955R1

Dear Dr. Ha Thu Nguyen

We’re pleased to inform you that your manuscript has been judged scientifically suitable for publication and will be formally accepted for publication once it meets all outstanding technical requirements.

Kind regards,

Biniyam Tefera Deressa

Academic Editor

PLOS ONE

Additional Editor Comments (optional):

Reviewers' comments:

Reviewer's Responses to Questions

**Comments to the Author**

1. If the authors have adequately addressed your comments raised in a previous round of review and you feel that this manuscript is now acceptable for publication, you may indicate that here to bypass the “Comments to the Author” section, enter your conflict of interest statement in the “Confidential to Editor” section, and submit your "Accept" recommendation.

Reviewer #1: All comments have been addressed

Reviewer #3: All comments have been addressed

2. Is the manuscript technically sound, and do the data support the conclusions?

Reviewer #1: Yes

Reviewer #3: Yes

3. Has the statistical analysis been performed appropriately and rigorously? 

Reviewer #1: Yes

Reviewer #3: Yes

4. Have the authors made all data underlying the findings in their manuscript fully available?

Reviewer #1: Yes

Reviewer #3: Yes

5. Is the manuscript presented in an intelligible fashion and written in standard English?

Reviewer #1: Yes

Reviewer #3: Yes

6. Review Comments to the Author

Reviewer #1: No further comments.

The authors have specifically address issues about the perspective adopted and further conducted value of information analysis

Reviewer #3: All the comments are addressed accordingly and results is acceptable. This article could play important role for use of trastuzumab in low middle income countries

7. PLOS authors have the option to publish the peer review history of their article (what does this mean?). If published, this will include your full peer review and any attached files.

Reviewer #1: **Yes: **Dr. Frida Ngalesoni

Reviewer #3: **Yes: **Dr Shah Zeb Khan

---

## [Editor Report · Acceptance letter]

7 Mar 2024

PONE-D-23-09955R1 

PLOS ONE

Dear Dr. Nguyen, 

I'm pleased to inform you that your manuscript has been deemed suitable for publication in PLOS ONE. Congratulations! Your manuscript is now being handed over to our production team.

Kind regards, 

on behalf of

Dr. Biniyam Tefera Deressa 

Academic Editor

PLOS ONE